# COVID-19 Pandemic Related Research in Africa: Bibliometric Analysis of Scholarly Output, Collaborations and Scientific Leadership

**DOI:** 10.3390/ijerph18147273

**Published:** 2021-07-07

**Authors:** Maxime Descartes Mbogning Fonkou, Nicola Luigi Bragazzi, Emmanuel Kagning Tsinda, Yagai Bouba, Gideon Sadikiel Mmbando, Jude Dzevela Kong

**Affiliations:** 1UFR IM2AG, Université Grenoble Alpes, 38000 Grenoble, France; dmbogning15@gmail.com; 2Laboratory for Industrial and Applied Mathematics (LIAM), Department of Mathematics and Statistics, York University, Toronto, ON M3J 1P3, Canada; robertobragazzi@gmail.com; 3Graduate School of Medicine, University of Tohoku, Sendai 980-8575, Japan; kagningemmanuel@gmail.com; 4Chantal BIYA International Reference Centre for Research on HIV/AIDS Prevention and Management (CIRCB), Yaoundé 3077, Cameroon; romeobouba@yahoo.fr; 5Graduate School of Life Sciences, University of Tohoku, Sendai 980-8577, Japan; gideonmmbando@gmail.com; 6Canadian Centre for Disease Modelling (CCDM), Department of Mathematics and Statistics, York University, Toronto, ON M3J 1P3, Canada

**Keywords:** bibliometrics, scientometrics, Africa, COVID-19 pandemic

## Abstract

Scientometrics enables scholars to assess and visualize emerging research trends and hot-spots in the scientific literature from a quantitative standpoint. In the last decades, Africa has nearly doubled its absolute count of scholarly output, even though its share in global knowledge production has dramatically decreased. The still-ongoing COVID-19 pandemic has profoundly impacted the way scholarly research is conducted, published, and disseminated. However, the COVID-19-related research focus, the scientific productivity, and the research collaborative network of African researchers during the ongoing COVID-19 pandemic remain to be elucidated. This study aimed to clarify the COVID-19 research patterns among African researchers and estimate the strength of collaborations and partnerships between African researchers and scholars from the rest of the world during the COVID-19 pandemic, collecting data from electronic scholarly databases such as Web of Science (WoS), PubMed/MEDLINE and African Journals OnLine (AJOL), the largest and prominent platform of African-published scholarly journals. We found that COVID-19-related collaboration patterns varied among African regions. For instance, most of the scholarly partnerships occurred with formerly colonial countries (such as European or North-American countries). In other cases, scholarly ties of North African countries were above all with the Kingdom of Saudi Arabia. In terms of number of publications, South Africa and Egypt were among the most productive countries. Bibliometrics and, in particular, scientometrics can help scholars identify research areas of particular interest, as well as emerging topics, such as the COVID-19 pandemic. With a specific focus on the still-ongoing viral outbreak, they can assist decision- and policy-makers in allocating funding and economic-financial, logistic, organizational, and human resources, based on the specific gaps and needs of a given country or research area.

## 1. Introduction

Scientometrics is emerging as a highly specialized branch of information science and as a sub-field of bibliometrics. It enables scholars to assess and visualize emerging research trends and hot-spots in the scientific literature from a quantitative standpoint. Moreover, scientometrics allows a rigorous analysis of patterns in terms of article usage and citations, generating an extensive series of measurements and indicators that can provide policy- and decision-makers with useful information concerning the effectiveness of their policies [1,2,3].

In the last several decades, Africa has nearly doubled its absolute count of scholarly output [4], even though its share in global knowledge production has dramatically decreased [5], with African countries losing approximately 11% of their share since their peak in 1987, and with Sub-Saharan Africa severely lagging behind, and reporting a loss of up to 31%. According to some updated statistics [6], African countries contribute to less than 1–1.5% of the global research output [7]. This limited contribution of African scholars to the global research output is in part impacted by the availability of adequate infrastructures and research collaborative networks.

The still ongoing “Coronavirus Disease 2019” (COVID-19) pandemic, caused by the emerging “Severe Acute Respiratory Syndrome-related Coronavirus type 2” (SARS-CoV-2), is an unprecedented infectious outbreak. Besides imposing a dramatic toll of cases and deaths, and being devastating both from a societal and economic-financial perspective, COVID-19 has profoundly impacted the way scholarly research is conducted, published and disseminated. Some authors [8] retrieved a pool of 441 articles relevant to the COVID-19 pandemic, approximately half of which (44.90%) were produced by mainland China, followed by the USA, Italy, Germany, and South Korea. Lower-middle-income and low-income countries contributed to 2.95%, and 0.23% of the output, respectively, with a negligible contribution from African countries and territories.

Bibliometric and scientometric analyses have been conducted to explore the emerging research focuses related to COVID-19. Such research focuses identified by researchers in mid-high-income countries include available treatment options, such as approved drugs or vaccines, or candidate management strategies [9,10,11]. While some bibliometric papers focus on summarizing research foci, other scientometric publications have assessed the scholarly output of researchers mainly from countries in Asia, America or Europe [12,13,14]. However, the COVID-19-related research focuses, the scientific productivity and the research collaborative network of African researchers during the ongoing COVID-19 pandemic remain to be elucidated.

Therefore, this study aimed to clarify the COVID-19 research patterns among African researchers and estimate the strength of collaborations and partnerships between African researchers and scholars from the rest of the world during the COVID-19 pandemic.

## 2. Materials and Methods

### 2.1. Bibliographic Search and Articles Identification

To identify the scientific literature on COVID-19 produced in Africa, we used a search string which consisted of terms related to COVID-19, the names of African countries and the main cities of these countries and territories (available at: https://github.com/descartesmbogning/How-the-COVID-19-pandemic-is-shaping-research-in-Africa-inequalities-in-scholarly-output-and-collab.git, accessed on 30 May 2021; Appendix A). Data was collected from electronic scholarly databases such as Web of Science (WoS), PubMed/MEDLINE and African Journals OnLine (AJOL), the largest and prominent platform of African-published scholarly journals. A database search was made on 12 March 2021 and publication date of papers was restricted to the period between 2019 and 2021. The number of records identified from PubMed/MEDLINE, WoS and AJOL were 4256, 5591 and 137, respectively. Figure 1 presents a flow-chart showing the selection process for the articles retained and analyzed.

### 2.2. Download of Bibliographic Information and Review of the Quality and Standardization of Data

Following the bibliographic search and document identification, we downloaded the data from the databases. After removing duplicates, 5704 articles were left and 5363 articles were included for downstream analyses after excluding the articles that did not match our inclusion criteria (341 articles). Duplicate removal was performed using ad hoc software (Endnote). The data file was then exported into a Microsoft Office Excel spreadsheet to count and exclude duplicated entries in some bibliographic fields. We found duplicated elements in institutional affiliations. We also reviewed and standardized entries of some fields. For example, among records from WoS, entries with a geographical origin that included “England”, “Scotland”, “Wales”, and “North Ireland” were renamed to “United Kingdom”.

### 2.3. Data Analyses

To analyze the COVID-19 publications from Africa, we grouped all countries according to World Bank geographical regions [15] and we assigned each country to its corresponding World Bank region. The World Bank regions are: East Asia and Pacific (EP), Europe and Central Asia (EC), Latin America & the Caribbean (LC), Middle East and North Africa (Middle East/North Africa) (MN), North America (NA), South Asia (SA), Sub-Saharan Africa (Eastern Africa/Southern Africa/Western Africa/Central Africa) (SSA).

Three types of analyses were considered to analyze the contribution of African scholars to COVID-19 literature.

As an introductory step to a better understanding of the global COVID-19 research, we quantified absolute scientific production by regions by counting the number of documents authored by researchers from each region. Moreover, we compared inter-regional, and international collaborations. We also compared the research leadership. The concepts used in the present study are defined as follows:

International collaboration: joint participation in the authorship of a document by researchers from two or more countries.

Inter-regional collaboration: joint participation in the authorship of a document by researchers from countries in two or more regions.

For each scientific publication, we list distinct authors’ institutional affiliations countries.

Geographical locations of authors were taken from authors’ institutional affiliations. The limitations section of this paper includes an in-depth explanation of shortcomings which should be considered when interpreting the results.

To specifically analyze COVID-19 research publications from African countries, we determined the number of documents authored by researchers from these countries. Furthermore, a direct collaboration network, representing the main African countries collaborating in COVID-19 research, was generated.

We analyzed the research subject areas and fields according to the disciplines that contributed the most to scientific production on COVID-19, as identified by means of the subject area classification of scientific journals in the WoS Core Collection (WoS-CC). To compare research orientations, we presented data for global research output, for publications produced solely by researchers from African countries, and publications produced through collaborations between researchers from African and non-African countries and territories.

Data analyses to extract publication indicators were performed using Excel and R [16]. Descriptive statistics (count, absolute and relative, as numbers and percentages) was performed.

Correlational analysis was conducted between variables of interests, for instance, between the strength of COVID-19 research collaboration networks between African and other institutions.

Correlation is a well-known bivariate analysis that determines the intensity of association and the direction of the relationship between two numerical variables. The value of the correlation coefficient varies between +1 and −1 in terms of the strength of the association. A value of 1 shows that the two quantitative variables are perfectly positively related. A value of −1 shows that the two quantitative variables are perfectly negatively related. There are two major types of correlation coefficients: the Pearson and the Spearman correlation coefficients. The latter is a nonparametric correlation coefficient, that should be used if one or more of the following conditions holds true: (i) at least one of the variables measured (x or y) is on an ordinal scale; (ii) neither x nor y is normally distributed; (iii) the sample size is small; and, (iv) the relationship is non-linear. Specifically, in the present bibliometric study, we did not use the Pearson’s correlation method because our variable of interests did not meet normality assumption. A number of published bibliometric reports used the Spearman’s correlation coefficient to measure the strength of relationship between variables of interest [17,18].

## 3. Results

### 3.1. African Scientific Production by Region and Degree of International Collaborations

Considering African participation in the scientific production related to COVID-19, Northern Africa and Southern Africa are the main contributors, with Northern Africa accounting for 34.07% of the total research output from Africa and Southern Africa accounting for 31.49% of the total output (Table 1). Together, these regions contributed up to 65.56% of the African scientific research production on COVID-19 that was indexed in the consulted sources. Central Africa contributed the least: only 5% of the African scientific production (Table 1). Amongst these scientific collaborations and partnerships, 41.21% of the scholarly research output was conducted by a country without collaboration with other African or non-African countries. This scientific production trend contrasts with the high percentages of collaborations observed in some specific African regions: namely, in Central Africa, 83.58% of the papers were published in collaboration with authors from more than one country, in Southern Africa 63.41%, and in Northern Africa, 57.22%.

Europe and Central Asia (EC) and North America (NA) based researchers are the main collaborators of African researchers, representing respectively 34.03% and 24.20% of scientific partnership contributing to the production related to COVID-19 (Table 1).

Northern Africa researchers collaborated in a marginal portion of their production with other African regions. Their main collaborators are from the Middle East and Europe & Central Asia (EC) researchers, with respectively 29.43% and 28.56% of scientific output related to COVID-19 (Table 1).

Within Africa, Central Africa researchers mostly collaborated with Western Africa (24.25%), followed by Southern Africa (22.01%). Outside Africa, we observed that Europe & Central Asia researchers were their principal collaborators (60.45%), followed by Northern America (30.97%) (Table 1).

Concerning Western Africa researchers, they mostly collaborated within Africa with Southern Africa (13.80%) and Eastern Africa (11.12%). Europe & Central Asia (38.13%) and Northern America (27.01%) researchers were their main collaborators outside of the continent (Table 1).

Southern Africa researchers mostly collaborated in Africa with Western Africa and Eastern Africa combined in less than 10% of their production. Europe & Central Asia researchers (42.33%), followed by Northern America (31.32%), were their principal collaborators outside Africa (Table 1).

Eastern Africa researchers collaborated with scholars residing in Africa mostly with Southern Africa (15.64%), followed by Western Africa (13.17%). Europe & Central Asia researchers (38.71%), followed by Northern America (31.09%), were their principal collaborators outside Africa (Table 1).

Figure 2 shows the strength of COVID-19 research collaboration networks between African and other institutions. The diameter of the circles and color codes represent the Spearman value of correlation coefficients. The larger (or the smaller) the value, the higher (or the lower) the collaboration strength between regions. The Figure shows a very weak correlation between researchers from Sub-Saharan Africa and Northern Africa.

### 3.2. Scientific Papers Published by Country and Degree of International Collaborations

Research production in Africa is concentrated in South Africa and Egypt, whose researchers contributed respectively to 27.07% and 22.75% of the articles from their regions. These countries are followed by Nigeria (14.12%), Morocco (6.82%), Ethiopia (6.00%) and Kenya (5.39%).

A total of fifty-two African countries contributed to Africa’s scientific production, with the number of articles by country ranging from 2 to 1452; the mean number of documents per country was 123.75 (std 276.68). In Central Africa, the country with the highest contribution was Cameroon with 127 (2.37%) documents, while Ethiopia led the production in Eastern Africa with 322 (6.00%) articles, Egypt in North Africa with 1220 (22.75%) documents, South Africa in Southern Africa with 1452 (27.07%) articles and Nigeria in Western Africa with 757 (14.12%) items (Table 2).

Among the most productive countries (>50 documents), Morocco, Ethiopia, Libya, Nigeria, and South Africa presented the lowest proportion of international collaborations. However, many other countries showed a value of international collaborations that exceeded 80% (Table 2).

### 3.3. African and Non-African Countries Collaboration and the Impact of Their Research

Appendix A contains data on the collaborations between researchers in Africa and those abroad. African research output on COVID-19 is characterized by its cooperative links, particularly with the USA and UK, which collaborated respectively with 49 and 45 African countries. We observed a significant number of links for colonial countries (Table 3 and Appendix A).

Concerning collaborations between African countries, South Africa stands out for its strong intra-regional ties, and it has become the main reference for research collaboration on COVID-19, both in Africa and among the top 20 most productive African countries. It has collaborated with 43 different African countries (Table 3 and Appendix A). Kenya ranks second in terms of collaborative leadership within Africa, followed by Nigeria and Cameroon which collaborated respectively with 40, 38 and 36 other African countries (Table 3 and Appendix A). On the other hand, Egypt was the second main contributor of scientific production, but it only collaborated with researchers of 28 African countries; it was, however, the main collaborator of Northern African countries. Egypt’s principal collaborator was Saudi Arabia, followed by the USA. It is also important to mention that Saudi Arabia was among the main collaborator of other North African countries (in particular, Arab speaking countries) (Table 3 and Appendix A).

The published articles considered in our analysis had an average citation per item of 4.15 and h-index of 57. These scores were higher in the scientific production in collaboration with non-African researchers, when compared to solely African collaboration, with a respective 5.7 vs. 2.2 for average citations per item and 53 vs. 28 for h-index (Table 3).

### 3.4. Active Journals

Pan African Medical Journal, South African Medical Journal, PLoS ONE, BMJ Global Health and Journal of Biomolecular Structure and Dynamics were the top five leading journals with, respectively, 246 (4.59%), 155 (2.89%), 97 (1.813%), 59 (1.10%) and 47 (0.88%) documents. In the list of top 15 active journals worldwide, two journals were in the field of microbiology and infections while the remaining were in the field of public health, environment, and general medicine (Table 4). The mean of impact factor of these top 15 journals was 6.26 with a standard deviation of 14.49 and median of 2.74.

Comparing the contribution of solely African researchers and those in collaboration with non-African researchers, the average impact factor of the top 15 journals was about six times higher in the group of researchers who collaborated with non-African researchers, at 10.09 versus 1.77, with medians of 3.20 versus 1.70.

### 3.5. Subject Areas Addressed in Publications on COVID-19 in Africa

The analysis on scientific COVID-19 output, produced by all countries worldwide, by African countries alone, and through Africa plus global collaborations, showed differences in terms of disciplinary orientations and research topics. In terms of disciplines involved, discordance was noted between global publications versus solely African publications (Table 5). There is also a certain degree of discordance between solely African publications and Africa plus global collaborations. In contrast, there is great affinity between global research output and output from Africa plus global collaborations. Of note, COVID-19 publications from Africa alone and from Africa plus global collaborations were dominated by papers in the field of “Public, Environmental & Occupational Health,” and “Infectious Diseases”, although the proportions are slightly higher from Africa plus global collaborations. The disciplines of “Medicine, General & Internal” and “Health Policy & Services” were also of great relevance in the publications from African countries alone (Table 5).

## 4. Discussion

In the present bibliometric study, we found that COVID-19-related collaboration patterns varied among African regions. For instance, most of the scholarly partnerships occurred with formerly colonial countries (such as European or North-American countries). In other cases, scholarly ties of North African countries were above all with the Kingdom of Saudi Arabia. In terms of number of publications, South Africa and Egypt were among the most productive countries.

Bibliometrics and, in particular, scientometrics can help scholars identify research areas of particular interest, as well as emerging topics. Moreover, they can assist decision- and policy-makers in allocating funding and economic-financial, logistic, organizational, and human resources, based on the specific gaps and needs of a given country or research area.

Several important initiatives such as the “Hinari Access to Research for Health Programme” (HINARI) established by the World Health Organization (WHO), involving the scientific community and major publishers, have granted developing countries, including Africa, access to biomedical and health-related scientific literature [19]. Recently, the “National Institutes of Health” (NIH) has set up an initiative, termed as UNITE, in order to “end structural racism and achieve racial equity in the biomedical research enterprise”. Despite these efforts, the contribution of African countries to global knowledge has decreased in the last several years in terms of their share.

Our findings are in line with the existing literature, showing regional differences at the African level. COVID-19 has further distorted and exacerbated some inequalities in publishing and collaborating: for instance, a study [20] explored public health-related investigations conducted by African scholars in the period 1991–2005. An increase in the number of investigations and international collaborations was reported by 382% and 45-67%, respectively. However, uneven statistics concerning publishing and collaborating trends could be detected, with major regional variations.

In the present study, we found that COVID-19-related publications were mainly focused on topics like “Public, Environmental & Occupational Health”, “Infectious Diseases”, “Medicine, General & Internal” and “Health Policy & Services”. This particular focus can be understood considering that the global burden of disease in African countries is mostly generated by communicable disorders, which can be prevented by implementing public health interventions. It is interesting that in these research topics and fields, African countries as well as other developing countries and territories have performed better with respect to developed countries [21].

As such, we can conclude that the effect of the COVID-19 pandemic is nuanced and complex, on the one hand amplifying already existing inequalities [22,23], and on the other hand paving the way for new opportunities and catalyzing new venues [24,25].

However, despite its strengths, including the methodological rigor, the transparency and reproducibility of the present study, as well as the extensive series of analyses conducted, and the number of electronic scholarly databases mined, this investigation suffers from a number of shortcomings that should be properly acknowledged. Gray literature (via Google Scholar) was not included, as well as other major databases such as Scopus.

## 5. Conclusions

In conclusion, the ongoing COVID-19 pandemic has exerted a subtle, complex impact on research and publishing patterns in African countries. On the one hand, it has distorted and even amplified existing inequalities and disparities in terms of the amount of scholarly output, share of global knowledge, and patterns of collaborations, due to the chronic lack of infrastructures, facilities and resources that plagues Africa. On the other hand, COVID-19 provided new opportunities for research collaborations, which contributed to generating novel international partners for academic exchanges, and research collaborations. Furthermore, COVID-19 enabled the identification of research fields in which African scholars can strengthen their scientific leadership.

## Figures and Tables

**Figure 1 ijerph-18-07273-f001:**
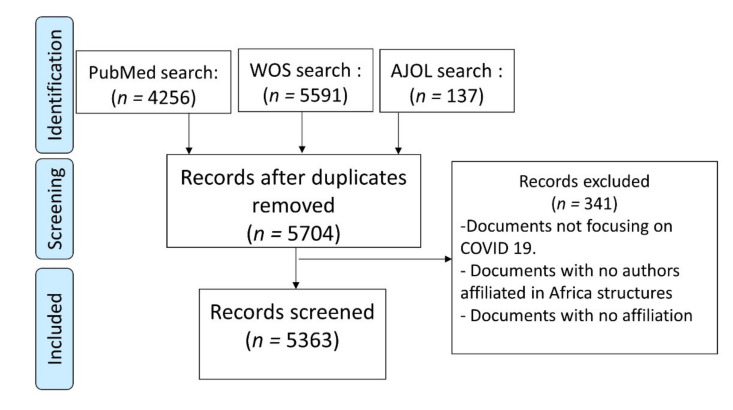
Flow-chart showing the selection process of articles included in the study.

**Figure 2 ijerph-18-07273-f002:**
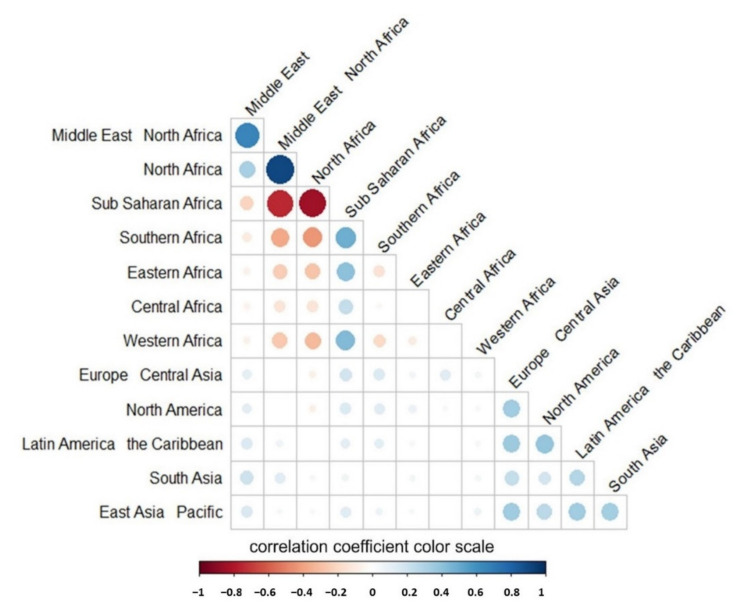
Correlation heatmap denoting the strength of COVID-19 research collaboration networks between African and other institutions. The diameter of the circles and color codes represent the Spearman value of correlation coefficients. The larger (or the lower) the value, the higher (or the lower) the collaboration strength between regions.

**Table 1 ijerph-18-07273-t001:** Scientific production on COVID-19, broken down by geographical region. *N* represents the number of articles and % the percentage.

Geographical Area	Articles		Eastern Africa		Southern Africa		Western Africa		Central Africa		Northern Africa	
	*N*	%	*N*	%	*N*	%	*N*	%	*N*	%	*N*	%
North America	1298	24.20	314	31.09	529	31.32	323	27.01	83	30.97	353	19.31
Latin America & the Caribbean	406	7.57	99	9.80	198	11.72	113	9.45	23	8.58	129	7.06
Europe & Central Asia	1825	34.03	391	38.71	715	42.33	456	38.13	162	60.45	522	28.56
East Asia & Pacific	868	16.19	188	18.61	337	19.95	252	21.07	47	17.54	250	13.68
Sub-Saharan Africa	3660	68.25	1010	100.00	1689	100.00	1196	100.00	268	100.00	126	6.90
Eastern Africa **	1010	18.83	1010	100.00	158	9.35	133	11.12	49	18.28	55	3.01
Southern Africa **	1689	31.49	158	15.64	1689	100.00	165	13.80	59	22.01	54	2.95
Western Africa **	1196	22.30	133	13.17	165	9.77	1196	100.00	65	24.25	63	3.45
Central Africa **	268	5.00	49	4.85	59	3.49	65	5.43	268	100.00	12	0.66
Middle East & North Africa	2068	38.56	124	12.28	179	10.60	155	12.96	24	8.96	1827	100.00
Middle East *	779	14.53	99	9.80	148	8.76	123	10.28	14	5.22	538	29.43
Northern Africa *	1827	34.07	55	5.45	54	3.20	63	5.27	12	4.48	1827	100.00
South Asia	471	8.78	122	12.08	157	9.30	134	11.20	26	9.70	183	10.01
Inter-regional collaboration	134	2.50	48	4.75	83	4.91	73	6.10	22	8.21	17	0.93
International collaboration	3019	56.29	617	61.09	988	58.50	697	58.28	202	75.37	1029	56.29
No or national collaboration	2210	41.21	345	34.16	618	36.59	426	35.62	44	16.42	782	42.78
Total	5363	100	1010	100.00	1689	100.00	1196	100.00	268	100.00	1828	100.00

** Sub-region of the Sub-Saharan Africa * Sub-region of the Middle East & North Africa.

**Table 2 ijerph-18-07273-t002:** Africa scientific production on COVID-19, by country. *N* represents the number of articles and % the percentage.

Country	World Bank Classifications by Region	Articles		No Collaboration		International Collaborations	
		*N*	%	*N*	%	*N*	%
South Africa	Southern Africa	1452	27.07	559	38.50	893	61.50
Egypt	North Africa	1220	22.75	405	33.20	815	66.80
Nigeria	Western Africa	757	14.12	310	40.95	447	59.05
Morocco	North Africa	366	6.82	248	67.76	118	32.24
Ethiopia	Eastern Africa	322	6.00	196	60.87	126	39.13
Kenya	Eastern Africa	289	5.39	53	18.34	236	81.66
Ghana	Western Africa	234	4.36	68	29.06	166	70.94
Uganda	Eastern Africa	169	3.15	31	18.34	138	81.66
Tunisia	North Africa	159	2.96	59	37.11	100	62.89
Cameroon	Central Africa	127	2.37	28	22.05	99	77.95
Algeria	North Africa	113	2.11	43	38.05	70	61.95
Sudan	Eastern Africa	113	2.11	30	26.55	83	73.45
Zimbabwe	Southern Africa	91	1.70	28	30.77	63	69.23
Tanzania	Eastern Africa	89	1.66	13	14.61	76	85.39
Senegal	Western Africa	88	1.64	20	22.73	68	77.27
D. R. Congo	Central Africa	81	1.51	11	13.58	70	86.42
Mozambique	Southern Africa	65	1.21	1	1.54	64	98.46
Malawi	Southern Africa	57	1.06	8	14.04	49	85.96
Zambia	Southern Africa	57	1.06	5	8.77	52	91.23
Libya	North Africa	56	1.04	27	48.21	29	51.79
Rwanda	Eastern Africa	51	0.95	4	7.84	47	92.16
Congo	Central Africa	43	0.80	3	6.98	40	93.02
Mali	Western Africa	41	0.76	3	7.32	38	92.68
Burkina Faso	Western Africa	35	0.65	9	25.71	26	74.29
Mauritius	Eastern Africa	32	0.60	10	31.25	22	68.75
Sierra Leone	Western Africa	31	0.58	3	9.68	28	90.32
Botswana	Southern Africa	29	0.54	9	31.03	20	68.97
Madagascar	Eastern Africa	29	0.54	6	20.69	23	79.31
Benin	Western Africa	27	0.50	3	11.11	24	88.89
The Gambia	Western Africa	23	0.43	4	17.39	19	82.61
Gabon	Central Africa	22	0.41	2	9.09	20	90.91
Guinea	Western Africa	21	0.39	3	14.29	18	85.71
Ivory Coast	Western Africa	21	0.39	0	0.00	21	100.00
Namibia	Southern Africa	17	0.32	6	35.29	11	64.71
Niger	Western Africa	15	0.28	3	20.00	12	80.00
Somalia	Eastern Africa	12	0.22	2	16.67	10	83.33
Swaziland	Southern Africa	11	0.21	5	45.45	6	54.55
Togo	Western Africa	10	0.19	0	0.00	10	100.00
Liberia	Western Africa	9	0.17	2	22.22	7	77.78
Guinea-Bissau	Western Africa	6	0.11	0	0.00	6	100.00
Mauritania	Western Africa	6	0.11	0	0.00	6	100.00
Burundi	Eastern Africa	5	0.09	0	0.00	5	100.00
Central African Republic	Central Africa	5	0.09	0	0.00	5	100.00
Chad	Central Africa	5	0.09	0	0.00	5	100.00
Eritrea	Eastern Africa	4	0.07	0	0.00	4	100.00
South Sudan	Eastern Africa	4	0.07	1	25.00	3	75.00
Angola	Southern Africa	3	0.06	3	100.00	0	0.00
Djibouti	North Africa	3	0.06	1	33.33	2	66.67
Equatorial Guinea	Central Africa	3	0.06	0	0.00	3	100.00
Lesotho	Southern Africa	3	0.06	1	33.33	2	66.67
Comoros	Eastern Africa	2	0.04	0	0.00	2	100.00
Seychelles	Eastern Africa	2	0.04	0	0.00	2	100.00

**Table 3 ijerph-18-07273-t003:** Collaboration and leadership of top 20 African countries in research papers on COVID-19.

			Total Collaborations						Collaborations with African Countries				Collaborations with Non-African Countries			
Rank	Country	Colonial Country	No. of Countries	No of Collaborations	% Collaborations	Average Citations Per Item	h-Index	Main Countries Collaborators (*n* Collaborations)	No. of Countries	Average Citations Per Item	h-Index	Main African Collaborators (n Collaborations)	No of Countries	Average Citations Per Item	h-Index	Main Non-African Collaborators (n Collaborations)
0	ALL	/	173	5363	100	4.15	57	South Africa (*n* = 1156); Egypt (*n* = 1220); USA (*n* = 1156)	48	2.2	28	South Africa (*n* = 636); Egypt (*n* = 413); Nigeria (*n* = 350)	119	5.7	53	USA (*n* = 1156); UK (*n* = 955); South Africa (*n* = 816)
1	South Africa	UK	138	893	61.50	5.80	33.00	USA (*n* = 414); UK (*n* = 360); Australia (*n* = 158)	43	2.42	13	Nigeria (*n* = 97); Kenya (*n* = 62); Ghana (*n* = 39)	95	8.53	33	USA (*n* = 414); UK (*n* = 360); Australia (*n* = 158)
2	Egypt	UK	128	815	66.80	4.89	31.00	Saudi Arabia (*n* = 317); USA (*n* = 257); UK (*n* = 172)	28	3.81	15	Nigeria (*n* = 36); South Africa (*n* = 35); Tunisia (*n* = 28)	100	5.43	27	Saudi Arabia (*n* = 317); USA (*n* = 257); UK (*n* = 172)
3	Nigeria	UK	125	447	59.05	3.93	19.00	UK (*n* = 175); USA (*n* = 159); South Africa (*n* = 97)	38	1.91	12	South Africa (*n* = 97); Egypt (*n* = 43); Egypt (*n* = 36)	87	5.59	17	UK (*n* = 175); USA (*n* = 159); India (*n* = 68)
4	Morocco	France	92	118	32.24	2.97	15.00	France (*n* = 33); USA (*n* = 32); Saudi Arabia (*n* = 24)	24	2.46	9	Egypt (*n* = 21); Algeria (*n* = 18); Tunisia (*n* = 12)	68	4.15	10	France (*n* = 33); USA (*n* = 32); Saudi Arabia (*n* = 24)
5	Ethiopia	/	77	126	39.13	3.94	16	USA (*n* = 48); UK (*n* = 32); India (*n* = 24)	28	2.76	10	Nigeria (*n* = 19); Kenya (*n* = 14); South Africa (*n* = 13)	49	6.29	12	USA (*n* = 48); UK (*n* = 32); India (*n* = 24)
6	Kenya	UK	116	236	81.66	3.89	15.00	USA (*n* = 114); UK (*n* = 98); Canada (*n* = 34)	40	2.2	6	South Africa (*n* = 62); Nigeria (*n* = 43); Uganda (*n* = 29)	76	4.38	13	USA (*n* = 114); UK (*n* = 98); Canada (*n* = 34)
7	Ghana	UK	97	166	70.94	2.92	12.00	UK (*n* = 68); USA (*n* = 60); South Africa (*n* = 39)	34	1.24	4	South Africa (*n* = 39); Nigeria (*n* = 32); Kenya (*n* = 19)	63	3.91	11	UK (*n* = 68); USA (*n* = 60); Germany (*n* = 29)
8	Uganda	UK	101	138	81.66	3.64	12.00	USA (*n* = 70); UK (*n* = 55); South Africa (*n* = 31)	29	3.66	4	South Africa (*n* = 31); Kenya (*n* = 29); Nigeria (*n* = 21)	72	3.63	12	USA (*n* = 70); UK (*n* = 55); Canada (*n* = 20)
9	Tunisia	France	89	100	62.89	6.03	16	USA (*n* = 39); Saudi Arabia (*n* = 33); Italy (*n* = 30)	17	2.43	4	Egypt (*n* = 28); Nigeria (*n* = 16); Morocco (*n* = 12)	72	8.2	16	USA (*n* = 39); Saudi Arabia (*n* = 33); Italy (*n* = 30)
10	Cameroon	UK/France	87	99	77.95	4.85	10.00	USA (*n* = 36); France (*n* = 33); UK (*n* = 27)	36	2.08	3	South Africa (*n* = 19); Kenya (*n* = 13); Ghana (*n* = 13)	51	6.35	9	USA (*n* = 36); France (*n* = 33); UK (*n* = 27)
11	Sudan	UK	78	83	73.45	5.42	9.00	UK (*n* = 32); Saudi Arabia (*n* = 32); Egypt (*n* = 19)	23	3	4	Egypt (*n* = 19); Nigeria (*n* = 11); South Africa (*n* = 11)	55	6.14	9	UK (*n* = 32); Saudi Arabia (*n* = 32); USA (*n* = 15)
12	Algeria	France	81	70	61.95	2.16	8.00	Saudi Arabia (*n* = 19); Egypt (*n* = 18); Morocco (*n* = 18)	15	1.78	4	Egypt (*n* = 18); Morocco (*n* = 18); Tunisia (*n* = 11)	66	2.42	7	Saudi Arabia (*n* = 19); France (*n* = 16); USA (*n* = 15)
13	Zimbabwe	UK	89	63	69.23	3.77	9.00	South Africa (*n* = 32); UK (*n* = 29); USA (*n* = 23)	28	1.86	4	South Africa (*n* = 32); Kenya (*n* = 11); Uganda (*n* = 10)	61	5.36	8	UK (*n* = 29); USA (*n* = 23); Canada (*n* = 8)
14	Tanzania	UK	80	76	85.39	4.91	11.00	UK (*n* = 32); USA (*n* = 29); South Africa (*n* = 17)	25	4.17	4	South Africa (*n* = 17); Uganda (*n* = 17); Nigeria (*n* = 14)	55	5.14	9	UK (*n* = 32); USA (*n* = 29); Australia (*n* = 10)
15	Senegal	France	68	68	77.27	14.24	13.00	USA (*n* = 28); France (*n* = 22); UK (*n* = 22)	30	0.46	2	South Africa (*n* = 14); Nigeria (*n* = 9); Cameroon (*n* = 8)	38	20.6	13	USA (*n* = 28); France (*n* = 22); UK (*n* = 22)
16	Democratic Republic of the Congo	Belgium	56	70	86.42	3.6	7.00	Belgium (*n* = 30); USA (*n* = 26); UK (*n* = 23)	27	2.08	4	South Africa (*n* = 19); Kenya (*n* = 9); Cameroon (*n* = 7)	29	3.96	7	Belgium (*n* = 30); USA (*n* = 26); UK (*n* = 23)
17	Mozambique	Portugal	85	64	98.46	16.29	10.00	UK (*n* = 29); Spain (*n* = 28); USA (*n* = 20)	24	0	0	South Africa (*n* = 11); Uganda (*n* = 8); Tanzania (*n* = 5)	61	16.64	10	UK (*n* = 29); Spain (*n* = 28); USA (*n* = 20)
18	Zambia	UK	77	52	91.23	4.32	6.00	USA (*n* = 30); UK (*n* = 22); South Africa (*n* = 13)	27	0.8	1	South Africa (*n* = 13); Kenya (*n* = 10); Uganda (*n* = 9)	50	4.74	6	USA (*n* = 30); UK (*n* = 22); China (*n* = 10)
19	Malawi	UK	61	49	85.96	4.60	10.00	UK (*n* = 31); South Africa (*n* = 15); USA (*n* = 14)	25	1	1	South Africa (*n* = 15); Kenya (*n* = 11); Nigeria (*n* = 8)	36	5.57	9	UK (*n* = 31); USA (*n* = 14); Sweden (*n* = 8)
20	Libya	Italy	67	29	51.79	3.06	6.00	UK (*n* = 18); Saudi Arabia (*n* = 10); Egypt (*n* = 9)	7	2.83	5	Egypt (*n* = 9); Nigeria (*n* = 5); Kenya (*n* = 4)	60	3.29	5	UK (*n* = 18); Saudi Arabia (*n* = 10); Italy (*n* = 8)

**Table 4 ijerph-18-07273-t004:** Top 15 active journals publishing research papers on COVID-19 in Africa. IF represents the impact factor of the journal and % the percentage.

	Global Publications				Solely African Publications				African + Global Collaborations			
Rank	Journal	No.	%	IF	Journal	No.	%	IF	Journal	No.	%	IF
1	Pan African Medical Journal	246	4.59	0.51	Pan African Medical Journal	181	7.72	0.51	Pan African Medical Journal	65	2.15	0.51
2	South African Medical Journal	155	2.89	1.70	South African Medical Journal	138	5.89	1.70	BMJ Global Health	53	1.76	4.28
3	PLoS ONE	97	1.81	2.74	PLoS ONE	51	2.18	2.74	PLoS ONE	46	1.52	2.74
4	BMJ Global Health	59	1.10	4.28	African Journal of Primary Health Care and Family Medicine	27	1.15	0.93	Lancet	40	1.32	60.39
5	Journal of Biomolecular Structure and Dynamics	47	0.88	3.55	Risk Management and Healthcare Policy	24	1.02	2.84	International Journal of Environmental Research and Public Health	38	1.26	2.85
6	Lancet	46	0.86	60.39	Egyptian Journal of Radiology and Nuclear Medicine	23	0.98	0.29	International Journal of Infectious Diseases	32	1.06	3.20
7	International Journal of Infectious Diseases	45	0.84	3.20	Journal of Biomolecular Structure and Dynamics	22	0.94	3.55	Journal of Global Health	26	0.86	2.90
8	Journal of Medical Virology	45	0.84	2.02	Medical Hypotheses	21	0.90	1.38	American Journal of Tropical Medicine and Hygiene	51	1.69	2.13
9	International Journal of Environmental Research and Public Health	40	0.75	2.85	Journal of Medical Virology	21	0.90	2.02	Journal of Biomolecular Structure and Dynamics	25	0.83	3.55
10	Medical Hypotheses	34	0.63	1.38	Infection and Drug Resistance	18	0.77	2.98	BMJ-British Medical Journal	17	0.56	30.31
11	American Journal of Tropical Medicine and Hygiene	64	1.19	2.13	HTS Teologiese Studies/Theological Studies	17	0.73	0.52	Journal of Medical Virology	24	0.79	2.02
12	South African Medical Journal	33	0.62	1.29	South African Journal of Science	17	0.73	1.70	Frontiers in Public Health	24	0.79	2.13
13	Journal of Global Health	32	0.60	2.90	Heliyon	16	0.68	1.86	Travel Medicine and Infectious Disease	22	0.73	4.59
14	Frontiers in Public Health	31	0.58	2.13	International Journal of Infectious Diseases	13	0.55	3.20	The Lancet Global Health	21	0.70	21.60
15	Risk Management and Healthcare Policy	30	0.56	2.84	Pharmacy Education	13	0.55	0.30	Clinical Infectious Diseases	18	0.60	8.31

**Table 5 ijerph-18-07273-t005:** COVID-19-related research papers broken down by Web of Science categories, according to African involvement. N represents the number of articles.

Rank	WoS Category	Global Publications		Solely African Publications		African + Global Collaborations	
		*N*	%	*N*	%	*N*	%
1	Public. Environmental & Occupational Health	1013	22.70	438	22.20	575	23.11
2	Infectious Diseases	547	12.26	207	10.49	340	13.67
3	Medicine. General & Internal	404	9.05	201	10.19	203	8.16
4	Health Care Sciences & Services	249	5.58	123	6.23	126	5.06
5	Pharmacology & Pharmacy	225	5.04	91	4.61	134	5.39
6	Biochemistry & Molecular Biology	174	3.90	62	3.14	112	4.50
7	Multidisciplinary Sciences	173	3.88	82	4.16	91	3.66
8	Immunology	163	3.65	58	2.94	105	4.22
9	Respiratory System	156	3.50	65	3.29	91	3.66
10	Environmental Sciences	148	3.32	41	2.08	107	4.30
11	Medicine, Research & Experimental	143	3.20	71	3.60	72	2.89
12	Microbiology	130	2.91	51	2.58	79	3.18
13	Virology	128	2.87	53	2.69	75	3.01
14	Pediatrics	99	2.22	42	2.13	57	2.29
15	Health Policy & Services	99	2.22	60	3.04	39	1.57
16	Clinical Neurology	95	2.13	34	1.72	61	2.45
17	Surgery	95	2.13	46	2.33	49	1.97
18	Tropical Medicine	81	1.82	29	1.47	52	2.09
19	Oncology	81	1.82	37	1.88	44	1.77
20	Psychiatry	79	1.77	35	1.77	44	1.77

## Data Availability

Data is available at: https://github.com/descartesmbogning/How-the-COVID-19-pandemic-is-shaping-research-in-Africa-inequalities-in-scholarly-output-and-collab.git, accessed on 30 May 2021, and in the manuscript.

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
