# Peer review of "COVID-19 Pandemic Related Research in Africa: Bibliometric Analysis of Scholarly Output, Collaborations and Scientific Leadership"

_ijerph, 2021, doi:10.3390/ijerph18147273_

Round 1

Reviewer 1 Report

The article is very well written. I have only three minor comments:

  1. The abstract should be shorted.
  2.  I don´t think that the sub-sub-sections in 2.3 are necessary, just skip the subheadings.
  3. My most important comment: How does the results compare to normal times? Maybe this can be discussed, I know not als analysis can be repeated, it would be two much, but does the cooperation behaviour (e.g.) changed due to the crisis?

Author Response

Dear reviewer, 

thanks a lot for your comments.

Abstract has been shortened, as requested.

Sub-sections have been removed and skipped, as recommended.

Comparison has been added in discussion, as requested.

Reviewer 2 Report

The pandemic period has dramatically changed the landscape of today's world both in terms of the private and professional lives of each of us. Not surprisingly, many researchers are now addressing COVID-19 in a bid to help mitigate the global impact of the spreading disease. However, I am unable to recommend this article in its present form for publication. Below I listed several weaknesses both in terms of methodology and research implementation.

1. The proposed title does not correspond to the scope of the research conducted. In order to study the influence of some factor in "shaping" some phenomenon, we would have to:
- first, identify the patterns occurring in the phenomenon under study (in your case scientific collaboration) before the existence of the factor (in your case a pandemic),
- then analyze the patterns present in the given phenomenon during the existence of the factor,
- then observe the changes in these patterns,
- and finally, prove (with some statistical level of confidence) that these changes were influenced by that factor and not by any other.

What this article is really portraying is not the shaping of scientific research in general, but research strictly specific to COVID-19. And based on the research conducted, these conclusions cannot be generalized to other branches of science (but the Authors do so).

Given the above, the title should definitely be changed.
Also, too general conclusions, not resulting from research conducted should be revised, e.g.:
-"In conclusion, the ongoing COVID-19 pandemic has exerting a subtle, complex impact on research and publishing patterns in African countries",
- "In the present bibliometric study, we found that COVID-19 related collaboration patterns varied among African regions, being shaped and driven by historical, social, cultural, linguistic, and even religious determinants." - Where do you research historical, social, cultural, linguistic, and religious issues?
- "in the period March-May 2020 has found gender-disparities" - Where do you research gender issues, that you write that coincide with your research? 
-" In addition, we found that inherited historical, societal, cultural, linguistic, and even religious habits shape the scientific research collaboration in Africa", and more
- in general, the entire Conclusions chapter should focus on presenting the results of research actually conducted in this article.

2. The authors do not describe the conduct of the research clearly enough and sometimes lose control of the execution of the experiment. 
Therefore, a lot of changes are needed to improve the article:
- reveal the search queries for each bibliographic database platform individually,
- if a github repository has already been used, upload there please the processed datasets at each stage of the research, i.e. raw results downloaded from each bibliographic database, files after data-cleaning, after deduplication, etc.,
- explain to readers how the numerical values were calculated - e.g. in the case of scientific cooperation, whether it was a full or a fractional counting; argue your choices,
- Fig.1 5890-341 ?= 5363 - There are many such errors throughout the article - that's why I argue that the Authors have problems with the control of the experiment or do not clearly describe what they are doing, 
- please explain to potential users, why have you chosen the Spearman correlation method to discover the collaboration phenomena between given regions; please provide examples of usage of this method in the field of bibliometrics studies for world respected scholarly papers (esp. for collaboration between individual regions, countries); give an example how you calculate it for e.g. North Africa - the Middle East North Africa and Sub Saharian Africa and North Africa; is in your opinion no difference between strong positive and strong negative correlation? if there is a difference explain it to readers, please.
Putting the scale of colors at the bottom axis is misleading in terms of good practices of data visualization - if the scale of color has to be part of this figure it should be a legend,
- revise your calculations in general, because there are lots of errors in tables.

3. Minor typos.
- [98] thousands separator notation is misleading

In conclusion, in my opinion, the article is far from what can be called scientific reliability and therefore I do not recommend it in its current form for publication. 

In my opinion, the article needs such major changes that it should be re-submitted after implementing all necessary improvements.

Author Response

Dear reviewer,

we agree with your observations and comments.

Title has been changed into COVID-19 Pandemic related Research in Africa: Bibliometric Analysis of Scholarly output,  Collaborations and Scientific Leadership, as you recommended.

We avoided generalizations to other disciplines. We revised conclusions (and abstract accordingly).

Search strings at the level of the individual database are now provided in the supplementary material. 

Every step of research, including pre-processing and processing, is now available in the github repository.

Typos have been amended and checked. 

Labels have been redone as suggested.

Spearman's correlation has been now explained, with references. 

Reviewer 3 Report

The introduction is well structured and developed, in provides relevant references that lead to the Materials and Methods section. The flow chart (fig1) with the numbers and references on github, makes the selection process transparent, the data cleaning process is well explained, as are the types of analyses and limitations and shortcomings.

The results are well presented, a nice study and interesting contribution.

In table 3, maybe enforce the line breaks (per country) for the presented collaboration numbers, for an improved readability, there is a lot of space in this table otherwise.

The discussion mentions religion as one of the determinants with regards to collaboration patterns. While the former colonial references have been provided, there is no reference that explicitly mentions the religion(s) dominant in a country and hence the reference is only implicit. The authors might want to improve this reference.

Reference 6 is not correctly cited; "some updated statistics", no year of the contribution (2018) is given, and no title as to where it appears "Elsevier Connect", by "By Charon Duermeijer, PhD, Mohamed Amir, and Lucia Schoombee.

Author Response

We have corrected the reference and improved the discussion.

Round 2

Reviewer 2 Report

thank you for the changes made in the article. Unfortunately, some
information still needs to be improved. In Table 1, the values
​​entered in the table (Total) do not sum up to the declared numbers.
Instead of 5363, it should be 5364; instead of 1827 it should be 1828;
instead of 1196 it should be 1197; instead of 100 it should be 100,02(!).
It would also be necessary to explain where these percentages came from
(general comment also to other tables), e.g. in Table 1 the number of
articles N and the percentage are given? Where do such values ​​come
from? For example, North America N = 1298, 24,20% (where does this
percentage come from? Percent of what?).
The article needs improvement. I cannot recommend the article for
publication in this form. 

Author Response

Thanks. 

We have checked again and corrected all typos and mistakes. We have clarified N and %. For example, NA publications are 1,298 (divided by 5,364 we got the percentage).